# Evaluation of the Recovery of Idiopathic Sudden Sensorineural Hearing Loss Based on Estimated Hearing Disorders

**Tadashi Nishimura** [1,*], **Tadao Okayasu** [1], **Chihiro Morimoto** [1], **Sakie Akasaka** [1], **Tadashi Kitahara** [1] **and Hiroshi Hosoi** [2]

1   Department of Otolaryngology-Head and Neck Surgery, Nara Medical University, 840 Shijo-cho, Kashihara 634-8522, Nara, Japan

2   MBT (Medicine-Based Town) Institute, Nara Medical University, 840 Shijo-cho, Kashihara 634-8522, Nara, Japan

*   Correspondence: t-nishim@naramed-u.ac.jp; Tel.: +81-744-22-3051

**Abstract:** Various prognostic factors for idiopathic sudden sensorineural hearing loss (SSNHL) have been reported. Hearing loss directly derived from idiopathic SSNHL is important for understanding underlying pathogenesis and outcomes. We assessed the usefulness of evaluating hearing loss and recovery of idiopathic SSNHL on the basis of estimated hearing loss. The study included 115 patients whose characteristics and outcomes of hearing loss were investigated. The effects of vertigo/dizziness and age on hearing thresholds before/after treatment, nonaffected ear threshold, estimated hearing loss, improvement of hearing loss, and estimated remaining hearing loss were investigated. Vertigo/dizziness was a significant prognostic factor for hearing. In vertigo/dizziness patients, significantly more severe hearing loss and poorer improvement of hearing loss were observed above 500 Hz and below 1000 Hz, respectively. Severe hearing disorder remained at all frequencies. Conversely, post-treatment thresholds were significantly higher in the older population ($\geq$65 years), although no differences in pretreatment thresholds were observed between the younger ($\leq$64 years) and older populations. However, on the basis of nonaffected ear thresholds, previously existing hearing loss could have influenced the outcome. Thus, comparison of hearing outcomes between affected and nonaffected ears is essential for understanding hearing loss and outcomes in idiopathic SSNHL cases with existing hearing disorder.

**Keywords:** sudden deafness; vertigo; dizziness; age; hearing improvement; acute hearing loss

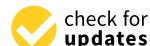



## 1. Introduction

Idiopathic sudden sensorineural hearing loss (SSNHL) is one of the major emergency diseases in otolaryngology. The frequency is not rare [1], and it is sometimes encountered in a clinical manner. Despite being a major disease, its pathogenesis and manifestation have not been completely elucidated, and no definite treatment has been established. Various prognostic factors for idiopathic SSNHL have been reported. The major factors related to the recovery of hearing loss are age [2–4], degree of hearing disorder [2,4], audiogram shape [2,5,6], symptom of vertigo/dizziness [4,7–9], and time from onset to treatment initiation [4,5,10]. Some of the possible risk factors for idiopathic SSNHL include older age, severe hearing disorder, down sloping audiogram, vertigo/dizziness, and delay of treatment initiation.

The outcomes of idiopathic SSNHL are usually evaluated according to the hearing thresholds before and after treatment [4–7]. Auditory ability depends on various factors and gradually declines owing to aging. The hearing disorders directly attributable to idiopathic SSNHL are not always the same despite similar thresholds before treatment. An increase in the hearing threshold obtained by subtracting the thresholds before disease onset from those before treatment directly indicates that the hearing disorder is derived from idiopathic SSNHL. Unfortunately, this disease occurs unpredictably and suddenly,

and the thresholds just before the onset are rarely recorded. Thus, the hearing loss directly induced by idiopathic SSNHL cannot be precisely determined.

In this study, we investigated the effects of prognostic factors according to hearing thresholds before treatment and improvement after treatment similar to previous studies [4–6]. In addition, the prognostic factors were investigated using the estimated hearing loss directly derived from idiopathic SSNHL. We assumed that the hearing thresholds of the affected ear before disease onset were similar to those of the nonaffected ear. Thus, the difference in hearing thresholds between the nonaffected and affected ears could help estimate the hearing disorder directly derived from idiopathic SSNHL. We investigated the prognostic factors on the basis of this estimated hearing loss. However, when both the affected and the nonaffected ears had existing hearing loss, the threshold in the nonaffected ear could not be regarded as the threshold in the affected ear before disease onset. After excluding these cases, we retrospectively investigated the prognostic factors for idiopathic SSNHL. This study aimed to assess the usefulness of evaluating the hearing loss and recovery of idiopathic SSNHL on the basis of estimated hearing loss.

## 2. Materials and Methods

Patients who visited our department for the treatment of idiopathic SSNHL from 2009 to 2017 were registered for this study. The criteria for enrollment into this study were as follows: (1) no ear diseases in both ears before the onset of idiopathic SSNHL; (2) no hearing disorder except presbycusis; (3) a time from onset of idiopathic SSNHL to treatment initiation within 14 days; (4) no radical treatment performed before visiting our department. The diagnosis of idiopathic SSNHL was performed according to the criteria defined by the Research Committee of the Ministry of Health and Welfare in Japan (Table 1). Occasionally, idiopathic SSNHL appears in both ears, and a nationwide epidemiological survey using the same diagnostic criteria reported the frequency of bilateral cases as approximately 1% [4]. In our department, no patients were diagnosed with bilateral idiopathic SSNHL during the study period. This study was approved by the ethics committee of Nara Medical University (No. 526). We provided participants with the opportunity to opt out, but no participants opted out of the study.

**Table 1.** Criteria for the diagnosis of idiopathic SSNHL.

| |
|---|
| Main Symptoms |
|     Sudden onset |
|     Sensorineural hearing loss, usually severe |
|     Unknown etiology |
| For reference |
|     Hearing loss (i.e., hearing loss of $\geq$30 dB over three consecutive frequencies) |
|     Sudden onset of hearing loss, but may progressively deteriorate over 72 h |
|     No history of recurrent episodes |
|     Unilateral hearing loss, but may be bilateral at the onset |
|     May be accompanied by tinnitus |
|     May be accompanied by vertigo, nausea, and/or vomiting, without recurrent episodes |
|     No cranial nerve symptoms other than those from cranial nerve VIII |
| Definite diagnosis, where all the above main symptoms are present |

Conventional pure tone audiometry was performed using an audiometer (AA-78, Rion; Kokubunji, Japan), and hearing thresholds were obtained at frequencies of 125, 250, 500, 1000, 2000, 4000, and 8000 Hz. The severity of hearing loss was evaluated according to the grading system established by the Research Committee of the Ministry of Health and Welfare in Japan. It was classified into four grades according to the audiometric mean of the five frequencies (250, 500, 1000, 2000, and 4000 Hz). The criteria of the audiometric means for Grades 1, 2, 3, and 4 were <40, 40–59, 60–89, and $\geq$90 dB, respectively [4]. The symptom of vertigo/dizziness was retrieved from the medical records and was considered

present when observed between disease onset and treatment initiation. The treatments are summarized in Table 2.

**Table 2.** The types of treatment.

| | |
|---|---|
| Systemic Corticosteroid with ATP, Vit B12 and PGE1 | |
| Intravenous administration (*N* = 97) | 97 (84.4%) |
| Systemic corticosteroid with ATP and Vit B12 | 9 (7.8%) |
| Intravenous administration (*N* = 4) | |
| Oral administration (*N* = 5) | |
| ATP, Vit B12, and PGE1 (No corticosteroids) | 6 (5.2%) |
| ATP and Vit B12 | 3 (2.6%) |

ATP: adenosine triphosphate; Vit B12: vitamin B12; PGE1: prostaglandin E1. Hydrocortisone sodium succinate and prednisolone were used for intravenous and oral administrations, respectively.

Hearing ability was monitored until participants recovered from the hearing loss. When the thresholds recovered to within 20 dB of the hearing level (HL) at all frequencies or reached the same level as the nonaffected ear, the hearing loss was considered completely recovered, and no further follow-up was performed. In the other cases, when the threshold remained stable for 1 month, the hearing loss was considered fixed. These final thresholds were regarded as the post-treatment thresholds. The treatment outcome of the hearing loss was diagnosed by calculating the audiometric mean of the five frequencies (250, 500, 1000, 2000, and 4000 Hz) according to the hearing improvement criteria defined by the Research Committee of the Ministry of Health and Welfare in Japan [4]. The outcome was categorized into "complete recovery", "marked improvement", "slight improvement", and "no change." When all the five frequencies in the final audiogram were ≤20 dB or improved to the same degree of hearing as the unaffected side, the outcome was diagnosed as "complete recovery". Audiometric mean improvements by ≥30, 10–29, and <10 dB were defined as "marked improvement", "slight improvement", and "no change", respectively.

The effect of age (≤64 or ≥65 years), sex, time from onset of idiopathic SSNHL to treatment initiation (within or after 7 days), and vertigo/dizziness on hearing loss grade were investigated. According to a previous study [4], hearing loss Grades 3 and 4 were defined as severe cases. In addition, the prognostic effects of several factors on idiopathic SSNHL were also investigated: age (≤64 or ≥65 years), sex, grade of hearing loss (Grades 1 and 2 or 3 and 4), time from disease onset to treatment initiation (within or after 7 days), and presence of vertigo/dizziness. According to a previous study [4], "slight improvement" and "no change" were defined as poor outcomes. Mann–Whitney's U test was used for statistical analysis.

In addition, we investigated the effects of vertigo/dizziness and age on the threshold at each frequency. For the statistical analysis, the threshold was regarded as the maxi-mum output level plus 5 dB when the participant could not hear the stimulus at the maximum output level. The respective maximum output levels at 125, 250, and 8000 Hz were set at 70, 90, and 100 dB HLs. The others were set at 110 dB HLs. In addition, assuming that the thresholds in the nonaffected ear were almost equal to those in the affected ear before the onset, the hearing loss directly derived from idiopathic SSNHL was estimated by subtracting the nonaffected ear threshold from the pretreatment threshold. The remaining hearing loss was also estimated by subtracting the nonaffected ear threshold from the post-treatment threshold. The improvement of the hearing loss was calculated by subtracting the post-treatment threshold from the pretreatment threshold. The effects of vertigo/dizziness (with or without) and age (≤64 or ≥65 years) on hearing disorder, improvement, and remaining hearing loss were also investigated at frequencies of 125–8000 Hz. The data were statistically analyzed by two-way analysis of variance (ANOVA) using SPSS ver. 22 (IBM Corporation, Armonk, NY, USA) at a 0.05 significance level. The Bonferroni method was used for post hoc comparisons.

## 3. Results

The participants' characteristics are presented in Table 3. A total of 66 and 49 participants were aged $\leq$ 64 (mean $\pm$ standard deviation, 47.8 $\pm$ 14.4) and $\geq$65 (73.3 $\pm$ 5.2) years, respectively. No obvious differences were observed with respect to sex and affected ear (right or left). In most patients, treatment was initiated within 7 days of disease onset. Vertigo/dizziness was identified in 38 patients. Regarding the severity of hearing loss, the percentages of Grades 1, 2, 3, and 4 were 9.6%, 20.0%, 45.2%, and 25.2%, respectively. After treatment, 38.3%, 19.1%, 25.2%, and 17.4% of the patients had "complete recovery", "marked improvement", "slight improvement", and "no change", respectively. The relationships between the patients' characteristics and grades of hearing loss are shown in Table 3. Vertigo/dizziness significantly influenced the hearing loss grade. Table 3 also shows the prognostic factors for hearing loss in idiopathic SSNHL. Vertigo/dizziness was also considered a significant prognostic factor.

**Table 3.** Characteristics of the included patients.

| | Number of Patients | Number of Grades 3 and 4 | Number of Poor Outcomes |
|---|---|---|---|
| All patients | 115 | 81 (70.4%) | 49 (42.6%) |
| Sex | | | |
|   Female | 58 (50.4%) | 38 (65.5%) | 20 (34.4%) |
|   Male | 57 (49.6%) | 43 (75.4%) | 25 (43.9%) |
| Affected ear | | | |
|   Left | 63 (54.8%) | 47 (74.6%) | 26 (41.2%) |
|   Right | 52 (45.2%) | 34 (65.4%) | 23 (44.2%) |
| Age | | | |
|   $\leq$64 years | 66 (57.4%) | 45 (68.2%) | 24 (36.4%) |
|   $\geq$65 years | 49 (42.6%) | 36 (73.5%) | 25 (51.0%) |
| Grade of hearing loss | | | |
|   Grades 1 and 2 | 34(29.6%) | | 15 (44.1%) |
|   Grades 3 and 4 | 81(70.4%) | | 34 (42.0%) |
| Time from the onset to start of treatment | | | |
|   $\leq$7 days | 98 (85.2%) | 67 (68.4%) | 41 (41.8%) |
|   8–14 days | 17 (14.8%) | 14 (82.4%) | 8 (47.1%) |
| Symptom of vertigo/dizziness | | | |
|   Positive | 38 (33.0%) | 32 (84.2%) * | 26 (68.4%) * |
|   Negative | 77 (67.0%) | 46 (59.7%) * | 23 (29.9%) * |

* Statistical significance ($p < 0.05$).

Figure 1 shows the hearing thresholds before and after treatment for the affected ear and for the nonaffected ear with and without vertigo/dizziness. The pretreatment thresholds for the patients with vertigo/dizziness were significantly higher than for those without vertigo/dizziness at 500–8000 Hz. Concerning the post-treatment thresholds, patients with vertigo/dizziness showed significantly higher thresholds at all frequencies. However, no differences in the thresholds of the nonaffected ears were observed between patients with and without vertigo/dizziness. The estimated hearing loss directly derived from the idiopathic SSNHL ear was significantly higher for patients with vertigo/dizziness at frequencies of 500–8000 Hz. In contrast, the improvements of hearing loss for patients with vertigo/dizziness were significantly lower at 125–1000 Hz. Notably, patients with vertigo/dizziness had significantly higher estimated remaining hearing loss at all frequencies.

Figure 2 shows the pre- and post-treatment thresholds of the affected ear and those of the nonaffected ear at ages $\leq$ 64 and $\geq$65 years. No differences in the thresholds were observed before treatment, and significantly higher thresholds were observed after treatment in the older population at all frequencies. Regarding the nonaffected ears, the thresholds in the older population were significantly higher than those in the younger population at all frequencies. The estimated hearing loss directly derived from idiopathic SSNHL was

significantly smaller for the older population than for the younger population at frequencies of 2000–8000 Hz. The improvements in the older population were significantly lower at 1000–8000 Hz. Concerning the estimated remaining hearing loss, no significant difference was found.

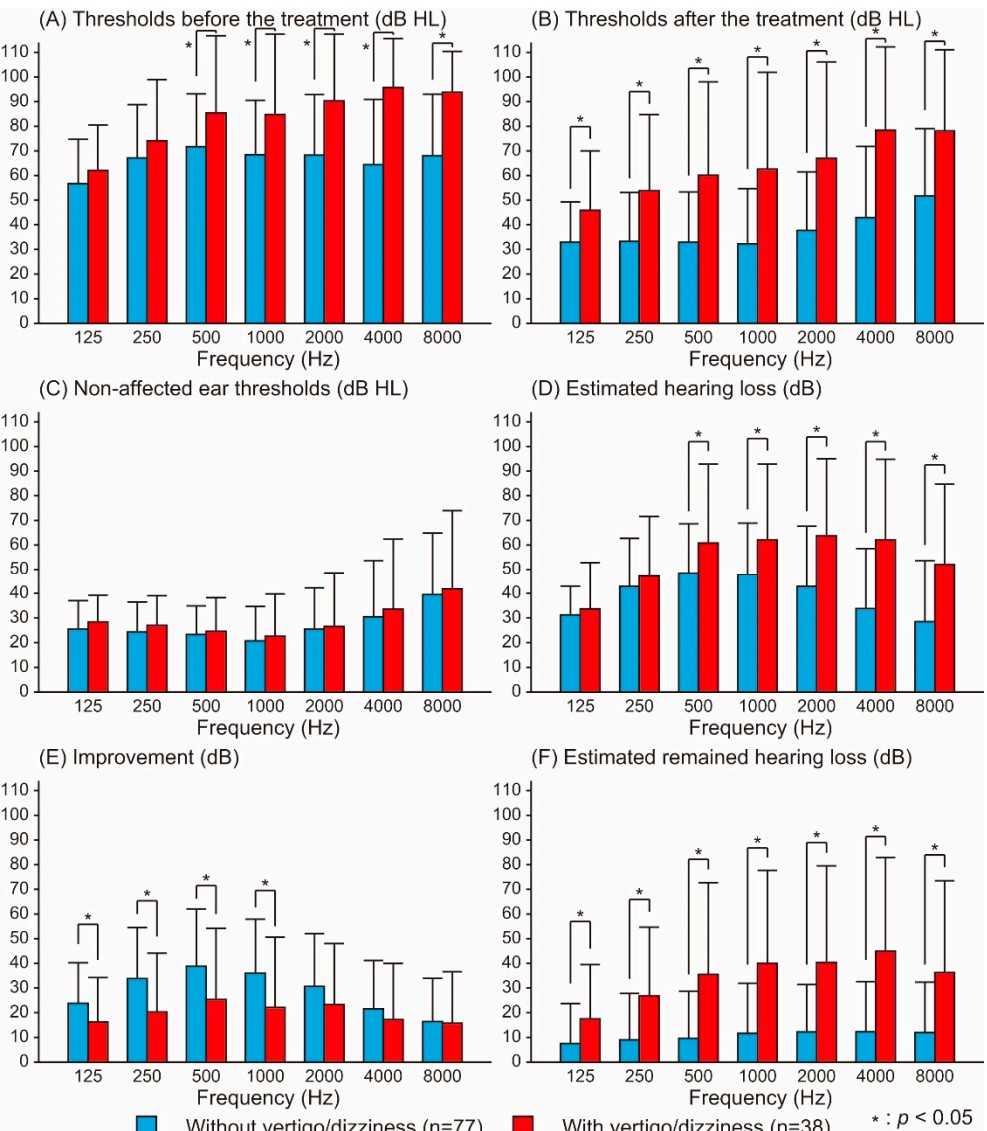

**Figure 1.** Comparison of audiometric characteristics between patients with and without vertigo/dizziness at each frequency. Figures show the thresholds before/after treatment (**A,B**), nonaffected ear thresholds (**C**), estimated hearing loss (**D**), improvement (**E**), and estimated remaining hearing loss (**F**). Two-way ANOVA revealed the significant main effects of the frequency and symptom of vertigo/dizziness, as well as the significant interaction between them, except for the comparison of nonaffected ear thresholds ($p < 0.05$). Vertical bars indicate the standard deviations.

Table 4 shows a comparison of the treatment types between patients with and without vertigo/dizziness and between the younger and older populations. No statistical significances were obtained by the chi-square for independence test.

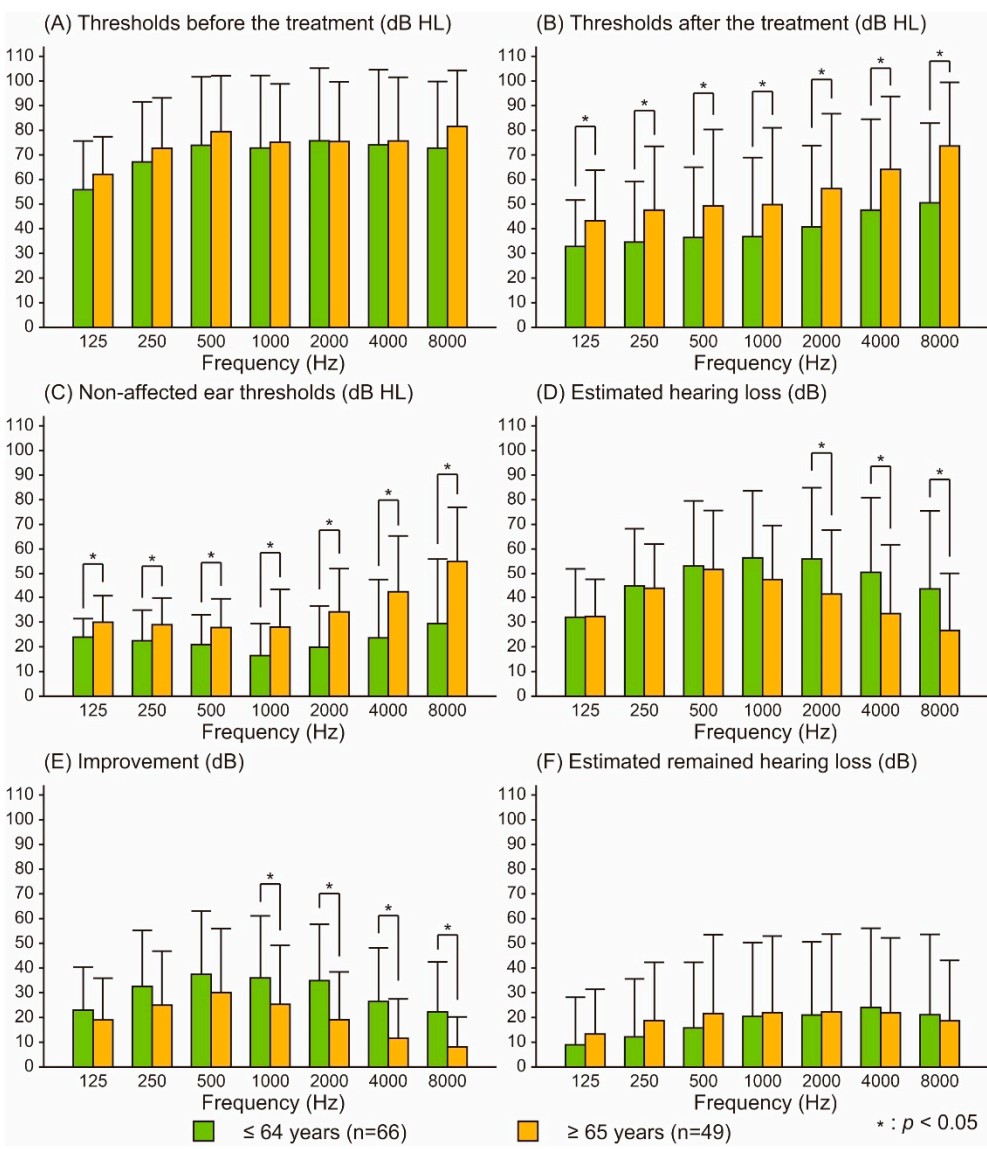

**Figure 2.** Comparison of audiometric characteristics between the patients aged ≤ 64 and ≥65 years at each frequency. Figures show the thresholds before/after treatment (**A**,**B**), nonaffected ear thresholds (**C**), estimated hearing loss (**D**), improvement (**E**), and estimated remaining hearing loss (**F**). Two-way ANOVA revealed significant main effects of the frequency and age, as well as a significant interaction between them, except for the comparison of thresholds before treatment and estimated remaining hearing loss ($p < 0.05$). Vertical bars indicate the standard deviations.

**Table 4.** The comparison of the types of the treatment.

| The Types of the Treatment | Vertigo/Dizziness | | Age | |
|---|---|---|---|---|
| | Without | With | ≤64 | ≥65 |
| Systemic corticosteroid with ATP, Vit B12, and PGE1 | | | | |
| Intravenous administration | 66 | 31 | 57 | 40 |
| Systemic corticosteroid with ATP and Vit B12 | | | | |
| Intravenous administration | 2 | 2 | 2 | 2 |
| Oral administration | 4 | 1 | 4 | 1 |
| ATP, Vit B12, and PGE1 (no corticosteroids) | 5 | 1 | 1 | 5 |
| ATP and Vit B12 | 0 | 3 | 2 | 1 |

## 4. Discussion

The hearing loss grade and treatment outcome in this study were evaluated according to the criteria defined by the Ministry of Health and Welfare in Japan and were compared to the results of a previous nationwide epidemiological survey that used the same criteria [4]. The hearing loss grade and treatment outcome in this study were almost similar to those of the previous study. In the present study, we found no differences in sex and the affected ear, consistent with the previous study. The proportion of patients aged ≥ 65 years was 42.6%, which was similar to the previous study (31.4%). In addition, the proportion of patients who received treatment within 7 days of disease onset was 85.2%, similar to the previous study (81.7%) [4]. When the period from disease onset to treatment initiation in the previous study was limited to within 14 days, the proportion of patients who received treatment within 7 days was 86.9%, which was also similar to that in the current study. Furthermore, no obvious differences in vertigo/dizziness and grade of hearing loss were found between the previous and present studies. Thus, the patients' characteristics in the present study are comparable to those of the general idiopathic SSNHL patients in Japan.

The treatment outcomes in the present study (complete recovery; marked improvement; slight improvement; no change) are similar to those in the previous study [4]. Regarding prognosis, vertigo/dizziness was considered a significant prognostic factor in the present study. In contrast, the previous study identified three other factors in addition to vertigo/dizziness: time from disease onset to treatment initiation, age, and hearing loss grade. The time from disease onset to treatment initiation may influence the result. Regarding age, the previous study stratified the patients into three age groups: ≤15, 16–64, and ≥65 years. The outcome in the ≥65 year age group was significantly worse than that in the 16–64 year age group [4]. The outcome in the ≤15 year age group was also worse than that in the 16–64 year age group, although not significantly. The present study, however, did not divide the patients aged ≤ 64 years into two groups because of the small sample size, which could have influenced the results. Some previous studies also reported that age is a significant prognostic factor [2–4], while others did not [10,11]. With respect to the hearing loss grade, the percentage of complete recovery and marked improvement in the mild hearing loss group (Grades 1 or 2) was 45%, significantly better than that in the severe hearing loss group (Grades 3 or 4) (38.3%). However, no significant difference was found in the present study (the respective percentages were 42% and 44%)

No differences were found in the thresholds of the nonaffected ear at all frequencies between patients with and without vertigo/dizziness, which suggests that hearing threshold before disease onset does not influence the symptom of vertigo/dizziness. In the affected ear, the thresholds before and after treatment for the patients with vertigo/dizziness were significantly higher than those for the patients without it at frequencies ≥ 500 Hz and all frequencies, respectively. These results suggest severe hearing loss and poor outcomes in patients with vertigo/dizziness, which agrees with the findings of previous studies [4,7–9]. In addition, for patients with vertigo/dizziness, middle to high frequencies were more sensitive. Previous studies reported that disequilibrium is associated with more severe forms of hearing loss [12], indicating a poorer prognosis, especially when accompanied by high-frequency hearing loss [11]. The extent of the inner ear lesion tends to correlate with the severity of cochlear damage, owing to the proximity of the basilar turn of the cochlea to the vestibular sense organ [12,13]. Ménière's disease is the representative disease with both hearing loss and vertigo/dizziness. There are various differences reported between Ménière's disease and idiopathic SSNHL, such as the recruitment phenomenon and abnormal vestibular-evoked myogenic potentials [14]. The hearing loss in Ménière's disease frequently occurs at a lower frequency range than that of idiopathic SSNHL. The pathophysiology of idiopathic SSNHL probably differs from that of Ménière's disease.

The estimated hearing loss directly derived from idiopathic SSNHL was significantly larger in patients with vertigo/dizziness, particularly by ≥20 dB at frequencies ≥ 2000 Hz. In contrast, regarding hearing improvement, no significant difference was found at high frequency between patients with and without vertigo/dizziness. This finding suggests

that the initial hearing loss at disease onset rather than poor improvement determines the remaining hearing disorder after treatment. Vertigo/dizziness is one of the poor prognostic factors for idiopathic SSNHL. Severe hearing loss at disease onset may result in poor outcomes in patients with vertigo/dizziness.

The hearing threshold after treatment was higher in the older population ($\geq$65), while no difference in pretreatment threshold was observed between the two age groups ($\leq$64 and $\geq$65). This result suggests a poorer outcome in the older population, which is consistent with previous findings that age may significantly influence prognosis [2–4]. However, a significant increase in hearing threshold was observed in the nonaffected ear, which could be attributed to presbycusis. In previous studies, the severity of hearing loss was evaluated on the basis of thresholds of the affected ear, which accordingly included the previously existing hearing loss. Furthermore, evaluation of the treatment outcome using the post-treatment threshold did not take into account the existing hearing loss caused by presbycusis. In the present result, the estimated hearing loss directly derived from idiopathic SSNHL and improvement in the older group were smaller and poorer than those in the younger population, respectively. Conversely, no significant difference in the estimated remaining hearing loss was found between the two age groups. According to the estimated hearing loss directly derived from idiopathic SSNHL, age was not always considered a prognostic factor, which is inconsistent with previous studies [2–4]. However, other previous studies did not indicate age as a significant prognosis factor [10,11]. The previously existing hearing loss might have influenced the results.

The findings of the present study suggest the usefulness of estimated hearing loss in evaluating idiopathic SSNHL and its recovery. However, quality data on hearing loss directly derived from idiopathic SSNHL are currently missing, thus warranting further research.

## 5. Conclusions

Vertigo/dizziness is a significant poor prognostic factor for hearing loss directly derived from idiopathic SSNHL. The improvement of the initial hearing loss does not significantly differ between patients with and without vertigo/dizziness. Substantial hearing loss occurs at disease onset and leads to poor treatment outcome. The treatment outcome is also poorer in the older population. Notably, the previously existing hearing loss may influence the treatment outcome of idiopathic SSNHL. The comparison of hearing outcomes between the affected and nonaffected ears is essential for understanding the hearing loss and outcomes in idiopathic SSNHL cases with existing hearing disorder.

**Author Contributions:** Conceptualization, T.N.; validation, T.O.; formal analysis, T.N. and T.O.; data curation, C.M. and S.A.; writing—original draft preparation, T.O.; writing—review and editing, T.N.; supervision, T.K. and H.H. All authors have read and agreed to the published version of the manuscript.

**Funding:** This research received no external funding.

**Institutional Review Board Statement:** The study was conducted in accordance with the Declaration of Helsinki and approved by the Ethics Committee of Nara Medical University (No. 526).

**Informed Consent Statement:** Patient consent was waived because the registered subjects had visited our hospital in the past, and have not visit any longer now. The information of the study was open to the public, and we provided subjects with the opportunity to opt out.

**Conflicts of Interest:** The authors declare no conflict of interest.

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
