# Peer review of "Evaluation of the Recovery of Idiopathic Sudden Sensorineural Hearing Loss Based on Estimated Hearing Disorders"

_audiolres, doi:10.3390/audiolres12050048_

Round 1

Reviewer 1 Report (Previous Reviewer 2)

I noticed that the authors took good notice of my suggestions and remarks in the previous review and changed everything that was necessary. The only thing is the English language, please, if possible, a little more than a minor spell check is required, e.g. with regard to the English style.

Author Response

Thank you for your comments and suggestions.

Reviewer 2 Report (Previous Reviewer 3)

The revised manuscript has fully resolved my concerns. I recommend it for publication.

Author Response

Thank you for your comments and suggestions.

This manuscript is a resubmission of an earlier submission. The following is a list of the peer review reports and author responses from that submission.

Round 1

Reviewer 1 Report

Dear authors,

Thank you very much for being able to review your manuscript.

After thorough reading and reviewing I do not think this manuscript is capable for publication. Let me give you some information and thoughts, which led me to this decision:

- Language and style needs distinct improvement, in many sections it is really hardto read and understand what you want to state, e.g. hearing loss vs. hearing rehabilitation

- Guidelines of the Ministry of Health and Welfare in Japan are not know by international readers, and are not provided. In order to allow for international comparion I would suggest to adapt to international known and accepted guidelinne, e.g. AAO-HNS Guidlines, just recently published and providing a great frameworlk in the field of hearing loss

- My major concern is the inadequate methodical set-up - concerning vertigo in dizziness. Even though Vertigo is retorspectivley concerned as a major factor for prognosis, the symptom was evaluated in this retrospective chart review only by potential mentioning of the symptom in the medical chart, affecting only 38/115 patients; a coherent conclusion can not be drawn. 

- finally, type of treetment provided among patients included is very diverse– a further factor not allowing for any conclusion on the whole group;

Reviewer 2 Report

See attached file.

Reviewer 3 Report

The authors performed a retrospective study of the prognosis factors in idiopathic SSNHL. A large group of patents were employed in this study and statistical analysis was performed to investigate the symptom of vertigo/dizziness and age on different threshold. In Discussion, the author also pointed out some limitations of this study. Overall, it was well designed. I have a few comments which may need to be resolved before publication.

Some comments:

1) Should the threshold be ‘higher’ or ‘lower’? What does ‘poorer’ mean?

2) The authors provided details about the treatment for the idiopathic SSNHL in Table. Can the authors clarify if the treatment for ≤ 64 years and ≥ 65 years) the same? Similarly, did patients with vertigo/dizziness require specific treatment?

3) At line 206-209, the author mentioned that the other study used three age groups while this study only used two. Can the authors provide the mean and standard deviation of the age for the ≤ 64 years and ≥ 65 years, respectively?

4) Can the author explain if one-way or two-way ANOVA was used? In Figure 1 and 2, there are two variables, which are the frequency and dizziness, and frequency and age.